# The Limited Effect of a History of COVID-19 on Antibody Titers and Adverse Reactions Following BNT162b2 Vaccination: A Single-Center Prospective Study

**DOI:** 10.3390/jcm11185388

**Published:** 2022-09-14

**Authors:** Hisako Kushima, Hiroshi Ishii, Yoshiaki Kinoshita, Yohei Koide, Yukie Komiya, Junko Kato, Mayumi Umehara, Miho Sakata, Motoyasu Miyazaki, Mikihiro Ikuta

**Affiliations:** 1Department of Infection Control and Prevention, Fukuoka University Chikushi Hospital, Chikushino 818-8502, Japan; 2Department of Respiratory Medicine, Fukuoka University Chikushi Hospital, Chikushino 818-8502, Japan; 3Department of Clinical Laboratory, Fukuoka University Chikushi Hospital, Chikushino 818-8502, Japan; 4Department of Pharmacy, Fukuoka University Chikushi Hospital, Chikushino 818-8502, Japan

**Keywords:** previous history of COVID-19, mRNA vaccine, anti-SARS-CoV-2 antibody

## Abstract

*Background and Objectives:* The need for, and ideal frequency of, the vaccination against coronavirus disease 2019 (COVID-19) of previously infected individuals have not yet been sufficiently evaluated. The aim of this study was to examine the anti-severe acute respiratory syndrome coronavirus 2 (SARS-CoV-2) antibody status and adverse reactions after vaccination among medical staff with or without a history of COVID-19. *Materials and Methods:* A single-center prospective study was performed at Fukuoka University Chikushi Hospital. We investigated the presence of the anti-SARS-CoV-2 antibody titer among medical staff before and after mRNA vaccination with the BNT162b2. The levels of immunoglobulin G antibody were quantitatively measured at six points—before vaccination, after the first vaccination, at three points after the second vaccination, and finally, after the third vaccination—and the levels were then compared based on the COVID-19 infection history. *Results:* The previously infected (before the first vaccination) subjects (*n* = 17) showed a marked increase in antibody titers two weeks after the first vaccination and four weeks after the second vaccination. Although they were able to maintain a certain level of antibody titers until 30 weeks after the second vaccination, the titers fell in the same way as observed in the non-infected subjects. The subjects who did not receive the third vaccination due to adverse reactions to previous vaccines (*n* = 1) or who were positive for COVID-19 prior to the third vaccination (*n* = 2) were excluded from the subsequent analyses. Among non-infected subjects (*n* = 36), smokers had lower peak antibody titers than the others. The previously infected subjects had a significantly higher incidence of adverse reactions after the first vaccination but had a similar incidence of adverse reactions after the second and third vaccinations compared to the non-infected subjects. *Conclusions:* A history of COVID-19 may influence only the initial increase in anti-SARS-CoV-2 antibody titers and the occurrence of adverse reactions after the first vaccination.

## 1. Introduction

Coronavirus disease 2019 (COVID-19) is a public health emergency and has rapidly spread worldwide since its first discovery in December 2019. COVID-19 has been reported in Japan since January 2020 and has had significant impacts on our daily lives and economic activities [1]. The mRNA vaccines developed against this background have been reported to be highly effective in preventing the onset and severity of COVID-19 with two to three or more vaccinations, and these vaccines have been recommended internationally as a key weapon in the fight against COVID-19 [2,3]. In Japan, the vaccination of healthcare workers began in February 2021, and the target population has gradually expanded. While previously infected individuals are believed to have also acquired immunity to the virus [4], the need for, and ideal frequency of, the vaccination of previously infected individuals have not yet been sufficiently evaluated [5,6].

In March 2021, the vaccination of medical staff in our hospital using the mRNA vaccine BNT162b2 (Pfizer Inc., New York, NY, USA) was started. The vaccine is usually administered twice via an intramuscular injection after a three-week interval, and a third booster is administered six to eight months after the second vaccination.

In this study, we evaluated the anti-severe acute respiratory syndrome coronavirus 2 (SARS-CoV-2) antibody status of medical staff before and after the first, second, and third vaccinations, in addition to the adverse reactions after each vaccination, according to whether or not they had a history of COVID-19.

## 2. Materials and Methods

### 2.1. Study Subjects

This was a prospective observational study conducted at Fukuoka University Chikushi Hospital in Fukuoka, Japan, between March 2021 and February 2022. The Ethics Committee of the Fukuoka University School of Medicine approved the study protocol (No. C21-12-001). Medical staff, mainly nurses and doctors, who gave their written informed consent regarding the blood sampling, were included in this study. They received the first vaccination against COVID-19 between March and April 2021 and then received the second vaccination three weeks later. The third vaccination was given about eight months after the second one. The subjects who did not receive the third vaccination were excluded from the subsequent analyses. Enrolled subjects were divided into two groups according to their history of COVID-19 before receiving the first vaccination. A non-infected subject was defined as a member of staff who had never been infected with COVID-19 or never had a positive screening test for SARS-CoV-2. All vaccines administered were the mRNA vaccine BNT162b2.

### 2.2. Data and Sample Collection

We examined the subjects’ background characteristics, including their age, sex, smoking status, and underlying diseases. In addition, a paper-based questionnaire survey was conducted regarding adverse reactions after each vaccination, including pain at the inoculation site, general malaise, headache, difficulty in daily work, fever, and medication for the adverse reactions. Blood samples were taken from the subjects before vaccination; 2 weeks after the first vaccination; around 4, 14, and 30 weeks after the second vaccination; and 4 weeks after the third vaccination.

### 2.3. Measurement of the Anti-SARS-CoV-2 Antibody Levels

The serum concentrations of anti-SARS-CoV-2 immunoglobulin G antibody were quantified using the HISCL^TM^ SARS-CoV-2 S-IgG (Sysmex, Kobe, Japan), according to the manufacturer’s instructions. This reagent for the anti-SARS-CoV-2 antibody is designed to detect immunoglobulin G acting against the SARS-CoV-2 spike protein S1 domain. To evaluate the associations between the subjects’ backgrounds or the development of adverse reactions and the peak of antibody titers (titers at 4 weeks after the second vaccination) or titer decline speed after the peak, each group was divided into two groups according to the median peak titer or median absolute value of the titer decline per day between 4 and 30 weeks after the second vaccination.

### 2.4. Statistical Analyses

Continuous data are shown as the group median (interquartile range), and categorical data are shown as the group number (percentage). The Mann–Whitney U-test was used to compare the non-parametric continuous variables between groups. For the comparison of the antibody titers, age-adjusted propensity score matching was used based on the baseline differences in age. Fisher’s exact test was used to compare the categorical variables. A *p* value of <0.05 was considered to indicate statistical significance. All statistical analyses were performed using R (version 3.2.2; R Foundation for Statistical Computing, Vienna, Austria).

## 3. Results

### 3.1. Subjects and Their Background Characteristics

A total of 53 participants, including 36 non-infected subjects and 17 subjects who had a history of COVID-19 between August 2020 and January 2021 (before receiving the first vaccination), were enrolled in this study. We were able to enroll all of the previously infected staff during the study period. All of them had been mildly ill or asymptomatic with COVID-19, including individuals who were positive for SARS-CoV-2 in the screening test at the time of the outbreak of a nosocomial cluster. No one received treatment for COVID-19, including neutralizing antibody agent. Three of the non-infected subjects did not receive the third vaccination due to adverse reactions to previous vaccines (*n* = 1) or because they were suffering COVID-19 prior to the third vaccination (*n* = 2). No subjects were excluded from the study due to job transfer or temporary retirement during the study period. None of the subjects were receiving systemic steroids or immunosuppressants due to underlying diseases. As shown in Table 1, the previously infected subjects were significantly younger than the non-infected subjects. There were no marked sex differences and no significant differences in smoking status or underlying diseases between the groups.

### 3.2. Anti-SARS-CoV-2 Antibody Titer

As shown in Table 2, compared to the non-infected subjects, previously infected subjects had significantly higher antibody titers not only at baseline but also two weeks after the first vaccination and at all points measured after the second vaccination. However, there was no significant difference in the level of antibody titers measured after the third vaccination between the groups, and the manner in which the antibody titers increased was similar across the groups. As shown in Table 3, the absolute values of the titer increase per day and titer decline per day after the peak level of the antibody titers among previously infected subjects were significantly higher than those among non-infected subjects.

When each group was divided into two groups according to the median absolute value of titer decline per day, the peak antibody titer in the fast-decline group [the median 9762 BAU/mL among previously infected subjects (*n* = 9) and 3285 BAU/mL among non-infected subjects (*n* = 15)] was significantly higher than the values for the slow-decline group [3056 BAU/mL among previously infected subjects (*n* = 8) and 1107 BAU/mL among non-infected subjects (*n* = 15), *p* < 0.001, respectively] (data not shown).In addition, in non-infected subjects, the low-peak-titer group (*n* = 15) and the slow-titer-decline group (*n* = 15) included more current smokers (*n* = 4) than the high-peak-titer group and fast-decline group (no smokers) (*p* = 0.042, respectively). Other the background factors did not affect the peak antibody titer or titer decline speed.

### 3.3. Adverse Reactions to BNT162b2

Table 4 shows the results of the questionnaire survey for adverse reactions after the first, second, and third vaccination. Although COVID-19 tended to induce a fever and the need for therapeutic agents for adverse reactions after the first vaccination, it did not affect the adverse reactions after the second or third vaccination.

The non-infected and fast-decline group for the antibody titers during the second half after the titer peak (*n* = 18) showed more adverse reactions just after the second vaccination than the non-infected and slow-decline group (*n* = 17), including pain at the inoculation site, headache, difficulty in daily work, a fever, and medication for adverse reactions (data not shown). However, there were no significant differences between these groups during the first half after the titer peak.

## 4. Discussion

In this study, we demonstrated the trends in anti-SARS-CoV-2 antibody titers after the triple vaccination of medical personnel, mainly nurses and doctors who did or did not have a history of COVID-19 before vaccination. Previously infected individuals showed a marked increase in antibody titers after the first and second vaccinations and were able to maintain a certain level of antibody titers for a period of time. However, the titers in the previously infected subjects fell in the same way as observed in the non-infected subjects. In addition, the antibody titers after the third vaccination increased to the same level in both groups, regardless of prior infection. In other words, if the subjects did not contract COVID-19 in the period leading up to the third vaccination, the level of antibody titers after the third vaccination increased regardless of whether or not the subjects had been infected prior to the first vaccination.

The absolute value of the decline in antibody titers per day after the peak titer increase among previously infected subjects was significantly higher than that among non-infected subjects. When each group was divided according to the speed of the decline in antibody titers after the second vaccination, the higher the peak value was, the faster the decline speed was. Although this may be a natural result, even subjects with high peak levels of antibody titers showed a definitive decline in antibody titers over time, regardless of their infection history. In terms of adverse reactions to BNT162b2, the previously infected subjects had a significantly higher incidence of adverse reactions after the first vaccination than non-infected subjects, but the rates of incidence of adverse reactions after the second and third vaccinations were similar to those among the uninfected subjects. Therefore, at least for participants in this study, a third vaccination might be recommended for both previously infected and non-infected individuals.

In this study, non-infected smokers had a lower peak antibody titer and a slower speed of titer decline than non-infected nonsmokers. A recent review [7] showed that current smokers had much lower levels of antibody titers or experienced a more rapid lowering of the vaccine-induced IgG than nonsmokers. This slightly different result may be attributed to the small number of subjects included in this study.

As reported in previous studies [8,9,10], previously infected subjects tended to have higher levels of antibody titers after vaccination than non-infected subjects. In one study, higher antibody titers were identified after the first vaccination among previously infected subjects, and these antibody titers exceeded the levels after the second vaccination among non-infected subjects. In addition, levels of antibody titers in previously infected subjects were even higher after the second vaccination than after the first vaccination. There is a report stating that previous COVID-19 infection appeared to elicit robust and sustained levels of SARS-CoV-2 antibodies in vaccinated individuals [10]. However, our study demonstrated that the levels of antibody titers in previously infected patients decreased steadily after the second vaccination, suggesting that repeated vaccination may be beneficial even for previously infected patients. As the precise antibody titer level required to prevent SARS-CoV-2 infection has not yet been determined, this is an issue to be explored in the future.

Several limitations associated with the present study warrant mention. One of the important limitations of the study was its small sample size at a single center. As only medical staff participated in this study, the average age of the population was relatively young, especially among the previously infected subjects. Therefore, it is unclear whether the same results would be obtained for the elderly. None of the previously infected subjects required oxygen during their illness, and all had been mildly ill or asymptomatic. It has been reported that the higher the severity of the disease is, the more antibodies are produced after infection [11]. Therefore, the present results are not representative of all previously infected patients. Although individuals newly infected during the study period were excluded, the possibility that asymptomatic, untested, infected individuals were included cannot be ruled out. Further studies including participants with various backgrounds are necessary. If the antibody titer required for the prevention of infection, disease onset, and severe disease, including the mutant strains, can be determined in future research, an antibody test will be useful not only for determining the effectiveness of vaccination but also for determining the need for routine vaccination and its intervals in the future.

## 5. Conclusions

In this study, in subjects previously affected by COVID-19, the antibody titer level tends to rise clearly at the beginning, and adverse reactions are likely to occur after the first vaccination, but the tendency thereafter is unclear. Multiple vaccinations might be recommended, regardless of one’s history of COVID-19.

## Figures and Tables

**Table 1 jcm-11-05388-t001:** Background characteristics of the participants.

Variables	Previously Infected Subjects (*n* = 17)	Non-Infected Subjects (*n* = 36)	*p* Value
Age, years	30 (24–40)	43.5 (39.5–50)	0.001
Sex, male	4 (23.5)	9 (25.0)	1
Smoking, %	1 (6.7)	6 (17.1)	0.659
Underlying disease			
Allergy, %	2 (13.3)	5 (14.3)	1
Malignancy, %	0 (0)	1 (2.9)	1
Hypertension, %	1 (6.7)	3 (8.6)	1
Dyslipidemia, %	1 (6.7)	4 (11.4)	1
Autoimmune disease, %	0 (0)	2 (5.7)	1
Diabetes mellitus, %	0 (0)	2 (5.7)	1
Asthma, %	0 (0)	3 (8.6)	0.542
Others, %	0 (0)	3 (8.6)	0.542

Values are expressed as medians (interquartile range) or number (%).

**Table 2 jcm-11-05388-t002:** Anti-SARS-CoV-2 antibody titers in medical staff.

Variables	Previously Infected Subjects (*n* = 17)	Non-Infected Subjects (*n* = 36)	*p* Value *
Before vaccination	63.0 (24.8–299)	0.0 (0.0–0.0)	<0.001
2 weeks after the first vaccination	3143 (2680–4415)	96.0 (49.7–184)	<0.001
4 weeks after the second vaccination	5615 (3104–9672)	2052 (1174–3225)	0.001
14 weeks after the second vaccination	1292 (992–1955)	526 (333–878)	0.004
30 weeks after the second vaccination	672 (468–1109)	194 (124–323)	0.001
4 weeks after the third vaccination	3956 (2997–6449)	4197 (2692–6066)	0.07

Values are expressed as medians (interquartile range) BAU/mL. * *p* value adjusted for age.

**Table 3 jcm-11-05388-t003:** Changes in titers per day before and after the peak of the antibody titers.

Variables	Previously Infected Subjects (*n* = 17)	Non-Infected Subjects (*n* = 36)	*p* Value **
Until the peak of titers *	129 (71.0–181)	42.6 (25.1–68.4)	0.002
Between 4 and 14 weeks after the second vaccination	−57.6 (−30.2–−101)	−21.2 (−12.3–−33.6)	0.004
Between 14 and 30 weeks after the second vaccination	−6.29 (−5.17–−8.42)	−3.89 (−1.99–−5.88)	0.041
Between 4 and 30 weeks after the second vaccination	−28.5 (−15.9–−48.1)	−11.1 (−6.76–−18.1)	0.003

Values are expressed as medians (interquartile range) BAU/mL. * The antibody titers at 4 weeks after the second vaccination. ** *p* value adjusted for age.

**Table 4 jcm-11-05388-t004:** Adverse reactions to BNT162b2.

Variables	Previously Infected Subjects (*n* = 17)	Non-Infected Subjects (*n* = 36)
After the first vaccination		
Pain at the inoculation site, %	13 (86.7)	25 (71.4)
General malaise, %	8 (53.3)	7 (20.0)
Headache, %	6 (40.0)	6 (17.1)
Difficulty in daily work, %	3 (20.0)	3 (8.6)
Fever, %	7 (46.7)	1 (2.9)
Medication for adverse reactions, %	7 (46.7)	2 (5.7)
After the second vaccination		
Pain at the inoculation site, %	9 (64.3)	24 (68.6)
General malaise, %	7 (50.0)	21 (60.0)
Headache, %	4 (28.6)	12 (34.3)
Difficulty in daily work, %	3 (21.4)	14 (40.0)
Fever, %	5 (35.7)	21 (60.0)
Medication for adverse reactions, %	5 (35.7)	13 (37.1)
After the third vaccination		
Pain at the inoculation site, %	10 (62.5)	21 (65.6)
General malaise, %	6 (37.5)	20 (62.5)
Headache, %	6 (37.5)	13 (40.6)
Difficulty in daily work, %	3 (18.8)	15 (46.9)
Fever, %	8 (50.0)	19 (59.4)
Medication for adverse reactions, %	7 (43.8)	15 (46.9)

Values are expressed as number (%).

## Data Availability

Not applicable.

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
