# Peer review of "The Limited Effect of a History of COVID-19 on Antibody Titers and Adverse Reactions Following BNT162b2 Vaccination: A Single-Center Prospective Study"

_jcm, 2022, doi:10.3390/jcm11185388_

Round 1
Reviewer 1 Report
The author studied about "Limited Effect of a History of COVID-19 to Antibody Titers and Adverse Reactions Following BNT162b2 Vaccination: A Single-Center Prospective Stud". This is an interesting observation.
The author mentioned in the manuscript that "adverse reactions" and also listed in Table 4. Did author measure any inflammatory markers to support this adverse reaction?
Author Response
We thank the reviewer for the thoughtful comment for us to revise our manuscript. The results of the adverse reactions after each vaccination were conducted by a paper-based questionnaire survey alone, and we could not measure any inflammatory marker. We have made some revisions to address the points raised by the other reviewer and would appreciate it if you could review our revised manuscript.
Reviewer 2 Report
To the authors,
Kushima et al. conducted a prospective study titled “Limited Effect of a History of COVID-19 to Antibody Titers and Adverse Reactions Following BNT162b2 Vaccination: A Single-Center Prospective Study”. The authors present an interesting manuscript about the correlation between previous Covid-19 infections and immunogenicity + reactogenicity following vaccination with the Pfizer/BNT vaccine. This is an important topic because many physicians wonder about the optimal timing for vaccinations after infections.
The article has the following strengths:
1. The methods appear to be sound.
2. The study population has a long follow-up period. I wonder, however, did you lose any patients in this period?
3. The manuscript is clear and concise.
4. The discussion is well balanced and some of the relevant limitations are addressed.
However, the article has also several shortcomings:
1. Abstract: Please define “history of Covid-19” Was there any time restriction? Only one episode of infection? Only history of infection before the first vaccination? How did you deal with subjects having an infection between the first and the third vaccination? This is partially explained in the text but should also be clearer in the abstract.
2. Abstract: Please provide actual numbers in your abstract. Reading only the abstract, statement like the following are difficult to understand: “The previously infected subjects (n=17) showed a rapid rise in antibody titer after the first vaccination but were not able to maintain antibody titers after the second vaccination.” Did they decline? If yes, when?
3. There was no formal sample size calculation, and the given sample size might be to low to infer strong conclusions. Could you provide some data on the statistical power of this study?
4. Please use the term sex instead of gender.
5. This is by nature not a randomized design. Because of this, confounder (eg, age!). Could you perform analyses in which you adjust for age (eg, multiple regression).
6. You present a lot of p values and I wonder if you should perform adjustment for multiple testing. In particular, testing for side effects may have produced false positives purely by chance.
Overall, the manuscript adds to the knowledge in this topic. However, my main concern is the limited sample size and the risk of confounding. These uncertainties should be expressed more clearly in the discussion of this study.
Round 2
Reviewer 2 Report
Thank you for responding to my concerns. No further comments form my side.